# Preservation of neurons in an AD 79 vitrified human brain

**Pierpaolo Petrone**[1]*, **Guido Giordano**[2], **Elena Vezzoli**[3], **Alessandra Pensa**[2], **Giuseppe Castaldo**[4], **Vincenzo Graziano**[1], **Francesco Sirano**[5], **Emanuele Capasso**[1], **Giuseppe Quaremba**[1,6], **Alessandro Vona**[2], **Maria Giuseppina Miano**[7], **Sergio Savino**[6], **Massimo Niola**[1]

**1** Dipartimento di Scienze Biomediche Avanzate, Università di Napoli Federico II, Naples, Italy, **2** Dipartimento di Scienze, Università degli Studi Roma Tre, Rome, Italy, **3** Dipartimento di Scienze Biomediche per la Salute, Università di Milano, Milan, Italy, **4** CEINGE Biotecnologie Avanzate Scarl, Naples, Italy, **5** Parco Archeologico di Ercolano, Ercolano, Naples, Italy, **6** Dipartimento di Ingegneria Industriale, Università di Napoli Federico II, Naples, Italy, **7** Istituto di Genetica e Biofisica "Adriano Buzzati-Traverso", Consiglio Nazionale delle Ricerche (CNR), Naples, Italy

* pipetron@unina.it

**Data Availability Statement:** All relevant data are within the manuscript and its Supporting Information files.

**Funding:** The author(s) received no specific funding for this work.

## Abstract

Detecting the ultrastructure of brain tissue in human archaeological remains is a rare event that can offer unique insights into the structure of the ancient central nervous system (CNS). Yet ancient brains reported in the literature show only poor preservation of neuronal structures. Using scanning electron microscopy (SEM) and advanced image processing tools, we describe the direct visualization of neuronal tissue in vitrified brain and spinal cord remains which we discovered in a male victim of the AD 79 eruption in Herculaneum. We show exceptionally well preserved ancient neurons from different regions of the human CNS at unprecedented resolution. This tissue typically consists of organic matter, as detected using energy-dispersive X-ray spectroscopy. By means of a self-developed neural image processing network, we also show specific details of the neuronal nanomorphology, like the typical myelin periodicity evidenced in the brain axons. The perfect state of preservation of these structures is due to the unique process of vitrification which occurred at Herculaneum. The discovery of proteins whose genes are expressed in the different region of the human adult brain further agree with the neuronal origin of the unusual archaeological find. The conversion of human tissue into glass is the result of sudden exposure to scorching volcanic ash and the concomitant rapid drop in temperature. The eruptive-induced process of natural vitrification, locking the cellular structure of the CNS, allowed us to study possibly the best known example in archaeology of extraordinarily well-preserved human neuronal tissue from the brain and spinal cord.

## Introduction

To date, there have been few discoveries of neuronal tissue from archaeological human remains [1]. Under certain taphonomic conditions that prevent soft tissue decomposition,

**Competing interests:** The authors have declared that no competing interests exist.

these brain remains are typically saponified [2]. However, ancient brains reported in the literature show only poor preservation of neuronal structures [3–8]. During our recent paleoforensic survey at the archaeological site of Herculaneum, we discovered glassy material within the cranial cavity of a human victim of the AD 79 Vesuvius eruption, apparently derived from the brain. Proteomics and mass spectrometry investigations of this material allowed us to identify several proteins of human brain origin and fatty acids of human hair fat, thus indicating preservation of vitrified human brain tissue [9].

Using scanning electron microscopy (SEM) and a specific image-processing tool based on a neural network, here we describe the unprecedented discovery of several typical central nervous system (CNS) ultrastructures from the victim's vitrified brain and spinal cord tissue. These remains are unique for the excellent quality of tissue preservation, giving us the opportunity to examine in detail the ultrastructure of a 2000-year-old human brain: due to a natural process of vitrification, at Herculaneum the CNS was "frozen" in its native condition, preserving intact remnant cell structures in the neuronal tissue [9]. The conversion of human tissue to glass (vitrification) occurred as a result of the rapid cooling of the volcanic ash deposit after exposure to the hot ash cloud at a temperature of about 500˚C [10–12]. Previous heating bone experiments showed analogous temperatures [13] that were also confirmed by our recent reflectance analysis on carbonized wood from Herculaneum [9].

## Background

The explosive eruption of Vesuvius volcano in AD 79 produced hot pyroclastic flows that hit and buried towns and settlements up to 20 kilometers away from the vent, causing thousands of fatalities [14–17]. Herculaneum, one of the main cities near the volcano, was buried beneath 20 meters thick pyroclastic flow deposits. Stratigraphy and paleotemperature determinations show that pyroclastic surges and flows reached the town with discrete pulses [10, 11]. Early arrivals severely damaged buildings and were mixed with debris and surface water, leading to highly variable local conditions of emplacement temperatures up to ca 500˚C [9], while later pyroclastic flow deposits buried the town at lower temperatures, on average at about 350˚C [10, 11].

The burial conditions guaranteed the complete preservation of this Roman city until the first discovery of its theatre in 1710. On 18 November 1739, during the Bourbon exploration, the first human victims were discovered [18]. A whole urban settlement buried by volcanic ashes, was gradually revealed and returned in its integrity [19]. But most exceptional of all was the discovery of several hundred human victims, most of which were found in 12 waterfront chambers during archaeological excavations of the suburban area in the second half of the 20th century [14, 15, 20]. The volcanic deposits preserved intact for centuries the corpses of these victims as time capsules, a source of unique bio-anthropological and paleoforensic information [9, 14, 21, 22].

In the mid-1960s, during the archaeological excavations of the *Collegium Augustalium*, a building dedicated to the cult of the Emperor Augustus, a victim was found lying on a wooden bed under a pile of volcanic ash [23]. Given the context of the discovery, a small service room, he is believed to have been the guardian of the College. The building is located at the corner of *Cardo III* where it intersects with Herculaneum's main street, the *Decumanus Maximus*. The College was the headquarters of the *Sodales Augustales*, made up of the most important representatives of the town [24]. The victim, a young adult male approximately 20 years of age, was lying ventrally, face down in the ash (Fig 1). The skull and postcranial skeletal remains show complete charring and cracking as a result of exposure to the hot pyroclastic surge [25], a high-speed turbulent cloud rich in hot gases, ash, and steam [26]. In a recent survey at the

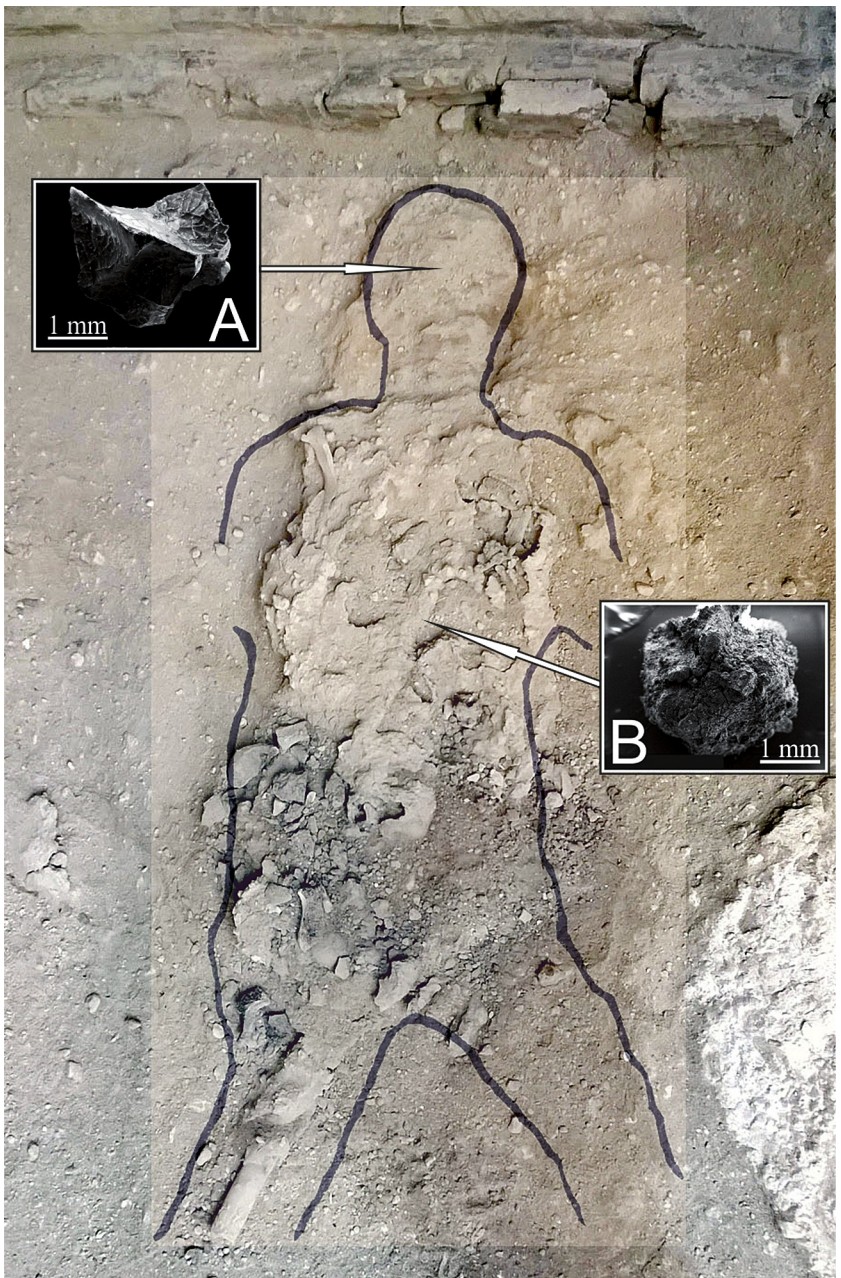

**Fig 1. 79 AD eruption victim (*Collegium Augustalium*, Herculaneum).** The body's features are outlined with the sketch drawn at the time of the discovery (1961). The posterior part of the skull (the occipital bone and part of the parietals) had completely exploded, leaving the inner part visible. A. Vitrified brain fragment collected from the inner part of the skull; B. Vitrified spinal cord fragment from the spine (SEM, scale bars in mm).

archaeological site, for the first time ever we discovered vitrified brain tissue inside the skull of this 79 AD eruption victim [9].

## Results and discussion

Here we report a unique piece of evidence regarding possibly the best preserved known example of CNS tissue in human archaeological remains. In order to detect the ultrastructural

architecture of these human remains, we carried out a scanning electron microscope (SEM) investigation to determine the nature and morphology of the vitrified tissue remains collected from the victim's skull (Fig 1A) and spine (Fig 1B). The samples being non-conductive, the SEM images were taken from the vitrified tissue as it was, so as not to compromise the integrity of this unique archaeological find. We found that this tissue consists exclusively of organic matter, given the high fraction of carbon (65·23 atom%, 57·69 mass%) and oxygen (31·01 atom%, 36·52 mass%) that we detected by energy-dispersive X-ray spectroscopy (EDS).

Notably, we show that the vitrified tissue consists of several ultrastructural features typical of the CNS (Fig 2). Based on our SEM investigation, a distinctive result is the preservation of a system of axon-like tubular structures running across the cerebral matrix (Fig 2A). The observed neuronal architecture is comparable to that detectable in biological samples after manual drop plunging protocol, or high-pressure freezing protocol followed by freeze-fracturing and cryo-SEM imaging [27–29]. The observed structures are elongated and round in shape, as described in the literature [30]. The mean diameter ranges between 550 and 830 nm, as expected for white matter axons which are in the medium 0·50–0·80 μm size range [31, 32]. Such tubular structures are smaller in diameter than blood vessels in the cerebrovascular system [33].

We also describe a complex neuronal system detected in a glassy residue of spinal cord. A basic structure consisting of cell bodies → axon-like is recurrent within the vitrified matrix (Fig 2B). The neuronal architecture observed via SEM shows the preservation of a number of cell bodies interconnected by a reticulum of tubular structures, whose morphology and size are analogous to those of neurons. Image processing was used to extract quantitative information from SEM images. The round-shaped cell bodies, mean size from 2·70 to 14·20 μm, show a cell membrane and an intracellular lumen filled by filamentous structures and nanovesicles (Fig 2B1 and 2B2). We classified the sections of the observed neuronal cell bodies in three groups, according to diameter (μm, mean ∓ SD) (14·17 ∓ 2·34; 8·06 ∓ 0·65; 2·72 ∓ 1·01) and mean area (μm$^2$, mean ∓ SD) (157·90 ∓ 51·20; 53·20 ∓ 13·67; 7·10 ∓ 3·90) (Table 1). The mean diameter (nm, mean ∓ SD) of 15 brain axons was 717·70 ∓ 93·10 and 672·00 ∓ 78·20 of 15 spinal cord axons (Table 2), values that are analogous to those of white matter axons [31, 32].

In addition, we show that the free axons identified in the vitrified brain tissue possess the typical myelin periodicity, as detected by our specific image processing tool based on a neural network process home-developed on Matlab® platform (S1 File) [34]. A region of interest (ROI) was identified before image acquisition. Geometrical measurements of the neuronal structures were performed through the segmentation of the ROI, after filtering the background by means of the active contour detection process [35]. In particular, in various brain axons (Fig 3A and 3B) we evidenced at least four different membrane layers wrapped around a single axon, thus forming compact myelin (Fig 3C). Thanks to this technique, it is possible to clearly distinguish a pattern in which lines of higher density are interspersed with intraperiod lines (Fig 3D), the same as seen *in vivo* for mammalian CNS myelin (Fig 3E) [36]. The denser lines represent the compacted cytoplasm, while the intraperiod lines are formed by close apposition of the membrane layers [37]. The results of our image processing procedure support the observation of the SEM images.

Unlike the myelinated axons observed in brain tissue, we show that axons in continuity with spinal cord body cells—as a rule, not myelinated—have a smaller diameter (nm, mean ∓ SD) (452·50 ∓ 20·50) (Fig 4A). Several evidence seems to confirm the neuronal origin of such structure: i. its diameter is significantly inferior to the minimal diameter of a capillary, the smallest of which in the human brain has a diameter ranging from 8 to 1 μm, and a wall thickness of about 1 μm [38, 39]; ii. it originates from the cellular membrane and it extends for

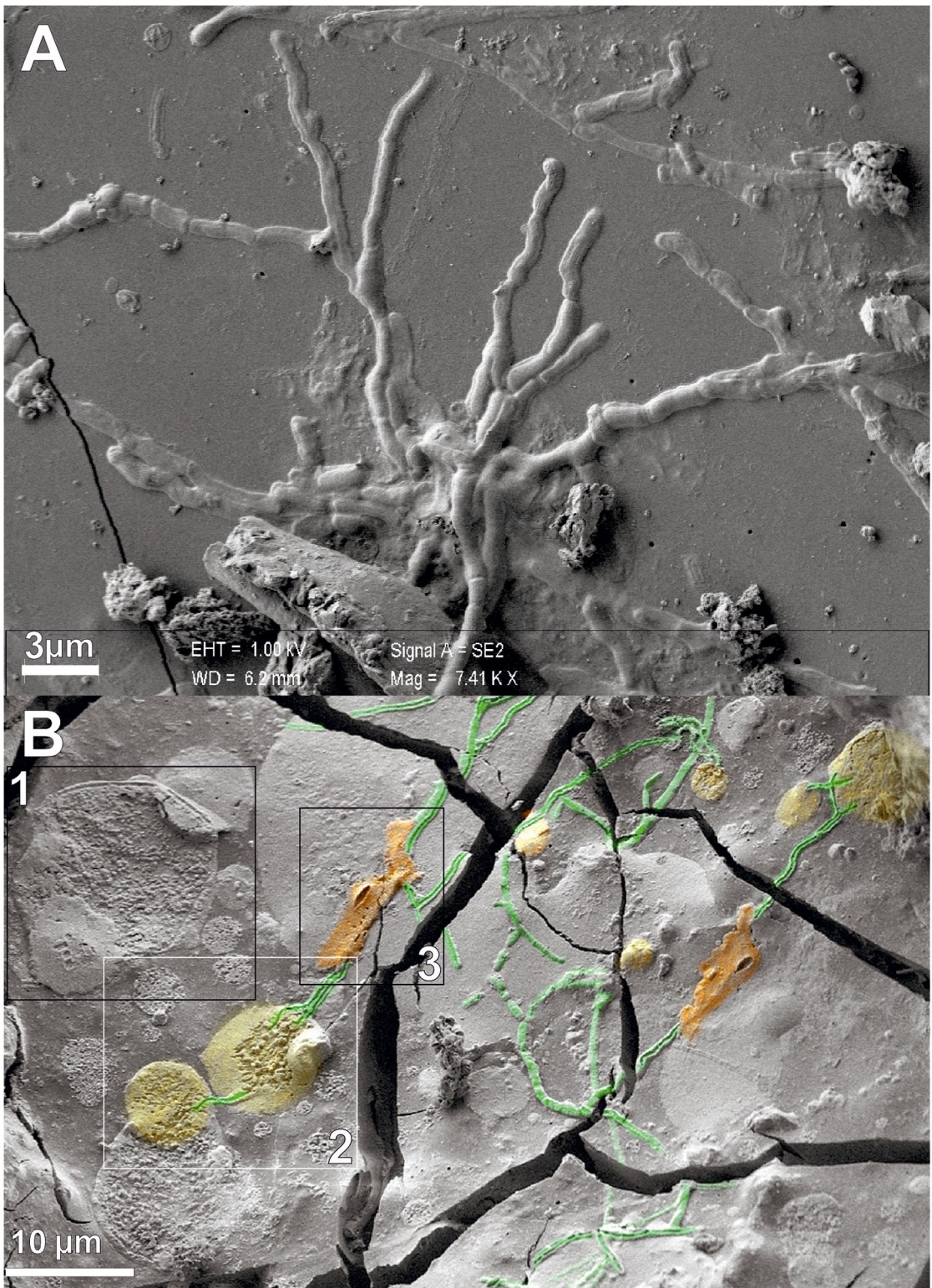

**Fig 2. Structures of the central nervous system.** A, SEM image of brain axons. B, SEM image of spinal cord axons (green) intercepting cell bodies and sheath-shaped structures (yellow and orange) (scale bars in micron).

**Table 1. Measures of diameter (μm) and mean area (μm²) of cells from the spinal cord fragment showed in Fig 1B.** A. Cell classification in three groups based on different size; B. Mean diameter and mean area of the three groups of cells.

| A | | |
|---|---|---|
| **1A group** | **diameter (μm)** | **mean area (μm²)** |
| 1 | 15,828 | 194,097 |
| 2 | 12,518 | 121,758 |
| **1B group** | **diameter (μm)** | **mean area (μm²)** |
| 1 | 8,641 | 70,431 |
| 2 | 8,106 | 51,446 |
| 3 | 8,346 | 53,770 |
| 4 | 7,146 | 37,265 |
| **1C group** | **diameter (μm)** | **mean area (μm²)** |
| 1 | 4,137 | 12,303 |
| 2 | 3,273 | 8,799 |
| 3 | 3,094 | 7,541 |
| 4 | 4,030 | 12,303 |
| 5 | 1,775 | 5,650 |
| 6 | 1,367 | 1,553 |
| 7 | 1,631 | 5,085 |
| 8 | 4,101 | 13,679 |
| 9 | 2,926 | 7,536 |
| 10 | 2,950 | 7,360 |
| 11 | 1,655 | 2,060 |
| 12 | 1,751 | 2,257 |
| 13 | 2,710 | 5,590 |

| B | | | |
|---|---|---|---|
| | **1A (mean ± SEM)** | **1B (mean ± SEM)** | **1C (mean ± SEM)** |
| **diameter (μm)** | 14,17 ± 1,655 | 8,06 ± 0,32 | 2,723 ± 0,279 |
| **mean area (μm²)** | 157,9 ± 36,17 | 53,23 ± 6,79 | 7,055 ± 1,100 |
| N° spinal cord fragments | 1 | 1 | 1 |
| Total N° round-shape cell body | 2 | 4 | 13 |

several microns away from the cell body; iii. its high similarity with the axon initial segment of a cerebral cortical pyramidal neuron [40]; iiii. the thickness of its wall (ca 70 nm) is far below the size of the wall thickness of a capillary (ca 1 to 0.5 μm) [38]. On this basis, we can assume that in our sample we have found a section of an axon initial segment which is a non-myelinated axon [41].

By our image processing procedure, we also identified the presence of regular tubular structures inside the cytoplasmic matrix of neuronal cell bodies (Fig 4B). The mean diameter of such nanostructures is approx. 23 nm, similarly to cytoplasmic microtubules [42]. In addition, the CNS matrix of both brain and spinal cord tissue appears to consist of recurring nanostructures with spiral morphology (Fig 4B). This evidence testifies that the vitrification-induced process of neuronal tissue preservation is the same for all the CNS structural components.

A further key aspect regards the CNS expression data of genes encoding a number of human proteins isolated from a sample of vitrified brain tissue, as previously reported by Petrone et al. [9]. By interrogating the Allen brain map (https://portal.brain-map.org/) and the Human Protein Atlas (HPA; www.proteinatlas.org) databases, we checked the expression of

**Table 2. Measures of mean diameter of the axons in brain and spinal cord.**

| Brain | diameter (nm) | Spinal cord | diameter (mm) |
|---|---|---|---|
| a.1 | 562 | a.1 | 552 |
| a.2 | 594 | a.2 | 590 |
| a.3 | 607 | a.3 | 605 |
| a.4 | 608 | a.4 | 610 |
| a.5 | 653 | a.5 | 629 |
| a.6 | 690 | a.6 | 643 |
| a.7 | 702 | a.7 | 643 |
| a.8 | 746 | a.8 | 650 |
| a.9 | 767 | a.9 | 661 |
| a.10 | 780 | a.10 | 680 |
| a.11 | 788 | a.11 | 712 |
| a.12 | 808 | a.12 | 719 |
| a.13 | 810 | a.13 | 765 |
| a.14 | 815 | a.14 | 791 |
| a.15 | 836 | a.15 | 831 |
| **mean ± SEM** | 717.7 ± 24.0 | **mean ± SEM** | 672.1 ± 20.2 |

the genes encoding those proteins identified in the vitrified tissue: Mediator complex subunit 13-like (*MED13L*; MIM 608661), 3-hydroxy-e-methylglutaryl-CoA reductase (*HMGCR*; MIM 142910), Kinesin family member 26B (*KIF26B*; MIM614026), WD repeat-containing protein 13 (*WDR13*; MIM 300512), SUPT6 interacting protein (*IWS1*), ATPase, H+ transporting liposomal V1 subunit F (*ATP6V1F*; MIM 607160) and Ribosomal protein S17 (*RPS17*; MIM18072).

HPA annotations obtained from the GTEx (Genotype-Tissue Expression project) human brain RNA-seq database revealed that all gene transcripts are present in the various parts of the human brain (cerebral cortex, basal ganglia, midbrain, pituitary gland, amygdala, cerebellum, hippocampus, hypothalamus and spinal cord), albeit differently expressed (Fig 5A and 5B) [43]. Many of them are exceptionally significant for neuronal functions since their mutations were detected in patients with brain pathologies. For example, *MED13L*, whose related protein is involved in neural differentiation [44], was found particularly abundant in adult cerebellum and its mutations were detected in patients with intellectual disability [45]. A further gene expressed more in the cerebellum than in other regions is *KIF26B*, whose mutations were found in patients with pontocerebellar hypoplasia [46]. It is involved in microtubule stabilization, which is indispensable for asymmetrical cell structure reorganization [47]. As shown in the HPA database, KIF26B co-localizes with microtubule and plasma membrane markers (Fig 6A and 6B) and is expressed in Purkinje cells and cells in granular layer (Fig 6C). Abundant in cerebellum and cortex is *HMGCR* encoding a cholesterol-regulating enzyme whose dosage was discovered upregulated in Alzheimer's disease patients [48].

Expressed in the brain but with a peak in cortex and cerebellum is *ATP6V1F*, encoding a subunit of the catalytic domain of the V-type H+-ATPase (Atp6v), a proton pump crucial for synaptic transmission that mediates the concentration of neurotransmitters into synaptic vesicles [49]. Interestingly, of the remaining three proteins isolated from the vitrified tissue, WDR13 is encoded by a highly conserved X-linked gene expressed in all brain regions, with a peak in the pituitary gland. There have been several associations of WDR13 with spatial memory and behavior in mice [50]. Similar to *WDR17*, an additional gene expressed more in the pituitary gland than in other regions is *RPS17*, encoding a ribosomal protein whose mutations

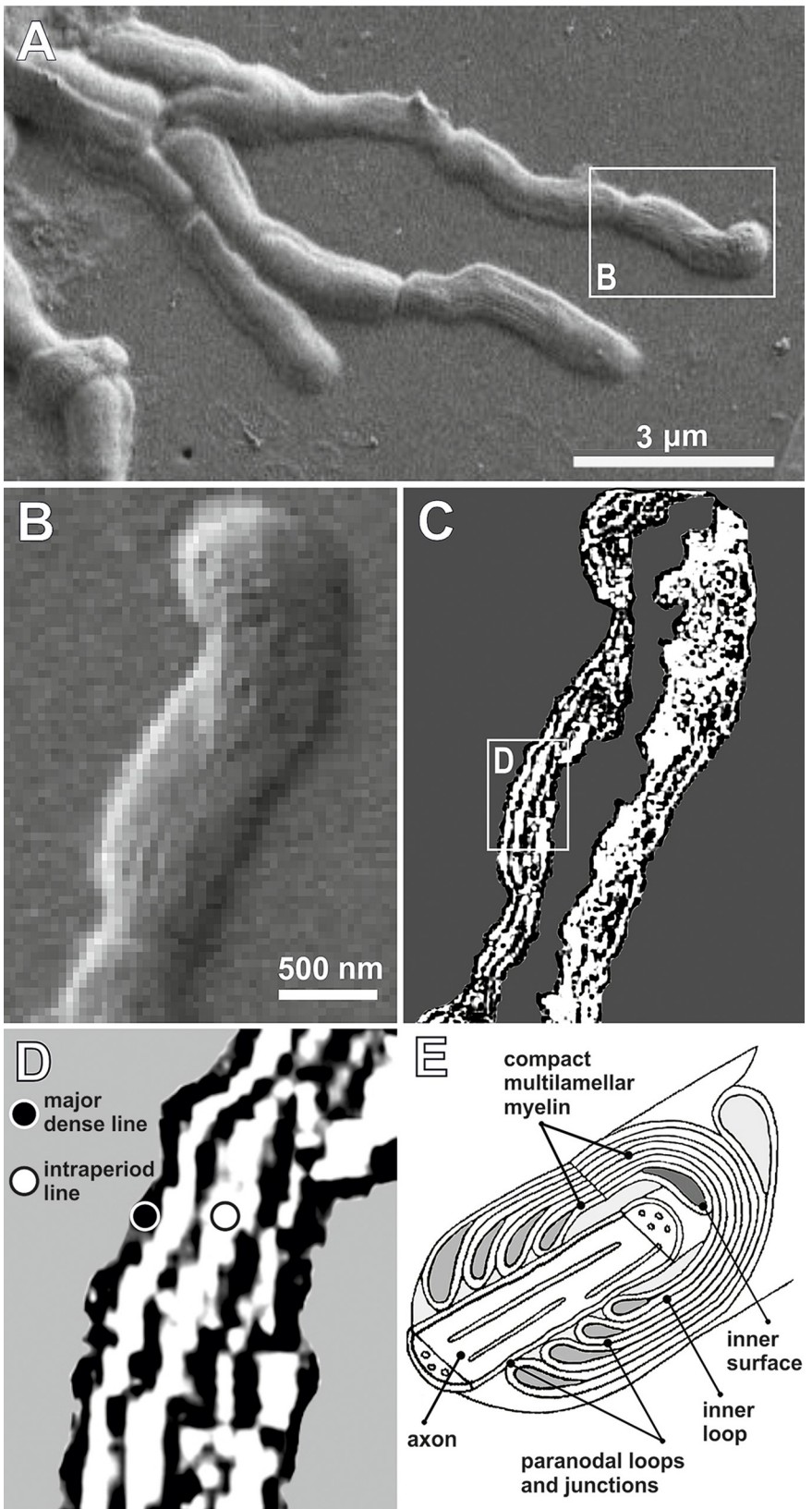

**Fig 3. Myelinated axon from the brain tissue.** A. Axon system and close view of a single axon unity (B) (SEM, scale bar in nm); C. Neural network elaboration of image B; D. Enlargement of image B (white box) showing several lamellae (a, b, c, d) of compact myelin; E. Schematic representation of a myelinated axon with compact multilamellar myelin [36].

do not alter neuronal functions but have been found in patients with Diamond-Blackfan anemia [51].

The detection of these proteins in vitrified brain tissue constitutes a valuable proof-of-concept supporting the data obtained *via* SEM. In particular, the finding of KIF26B involved in the organization of microtubules aligns well with the tubular structures identified inside the cytoplasmic matrix of cell bodies detected in the vitrified neuronal tissue (Fig 4B) [48, 49]. The notable discovery of ATP6V1F, essential for the synaptic vesicle cycle, constitutes a further clue in favor of the discovery of human neuronal cells. Moreover, the origin of the vitrified material from the posterior region of the skull (Fig 1A) further agrees with the discovery of proteins whose genes are mostly expressed in the cerebellum and spinal cord.

Based on the observed ultrastructural features, *i.e.*, morphological characteristics and morphometric measurements detected from the archaeological vitrified tissue, we classified cell bodies and axon-like processes as neurons and axons, respectively. We hypothesize that the unique natural process of vitrification occurring during the AD 79 eruption locked the structure of the CNS, thus preserving its morphology intact. This finding has an important implication concerning the specific environmental conditions that allowed the vitrification of the human brain and other neuronal tissues discovered at Herculaneum. Vitrification is a natural process that occurs when a liquid drops below its glass transition temperature, which depends largely on the cooling rate and the viscosity of the liquid [52, 53]. The preservation of this vitrified material implies that the brain was not destroyed during exposure to the hot pyroclastic flows and that time was allowed for its rapid cooling and transformation into glass before the final burial beneath further meters of hot pyroclastic debris. This indicates that some time-gaps must have occurred during the sequence of pyroclastic flow events that progressively hit and buried the town, as also recently suggested at Pompeii [54].

These results have important implications in the field of bioanthropological and volcanological research, which may open up a new line of biogeoarchaeological investigations on previously undetected evidence in the sites buried by the Vesuvius eruptions.

## Materials and methods

The human biological remains examined in this work were collected from the *Collegium Augustalium* at Herculaneum. Particular attention was paid to the detection of vitrified remains from the skull and spine. The samples were first observed with a 10x - 30x magnifying glass, and later analyzed and photographed by a stereo microscope (Leica M205 FA, magnification 8x - 160x, multidimensional stereo imaging system, CEINGE Biotecnologie Avanzate S. c.a.r.l.). All necessary permits were obtained for the study of the human specimen (Protocol 101/17, Ethics Committee for Biomedical Activities, AOU Federico II).

### Determination of sex and age

The specimen subject of this study is a young adult male. Sex and age at death were assessed according to standard diagnostic procedure. Given the fragmentation of the bone remains, only the best preserved diagnostic skull and postcranial features were considered for the attribution of sex [55–57]. Poor preservation of pelvis bones did not allow any reliable sexing

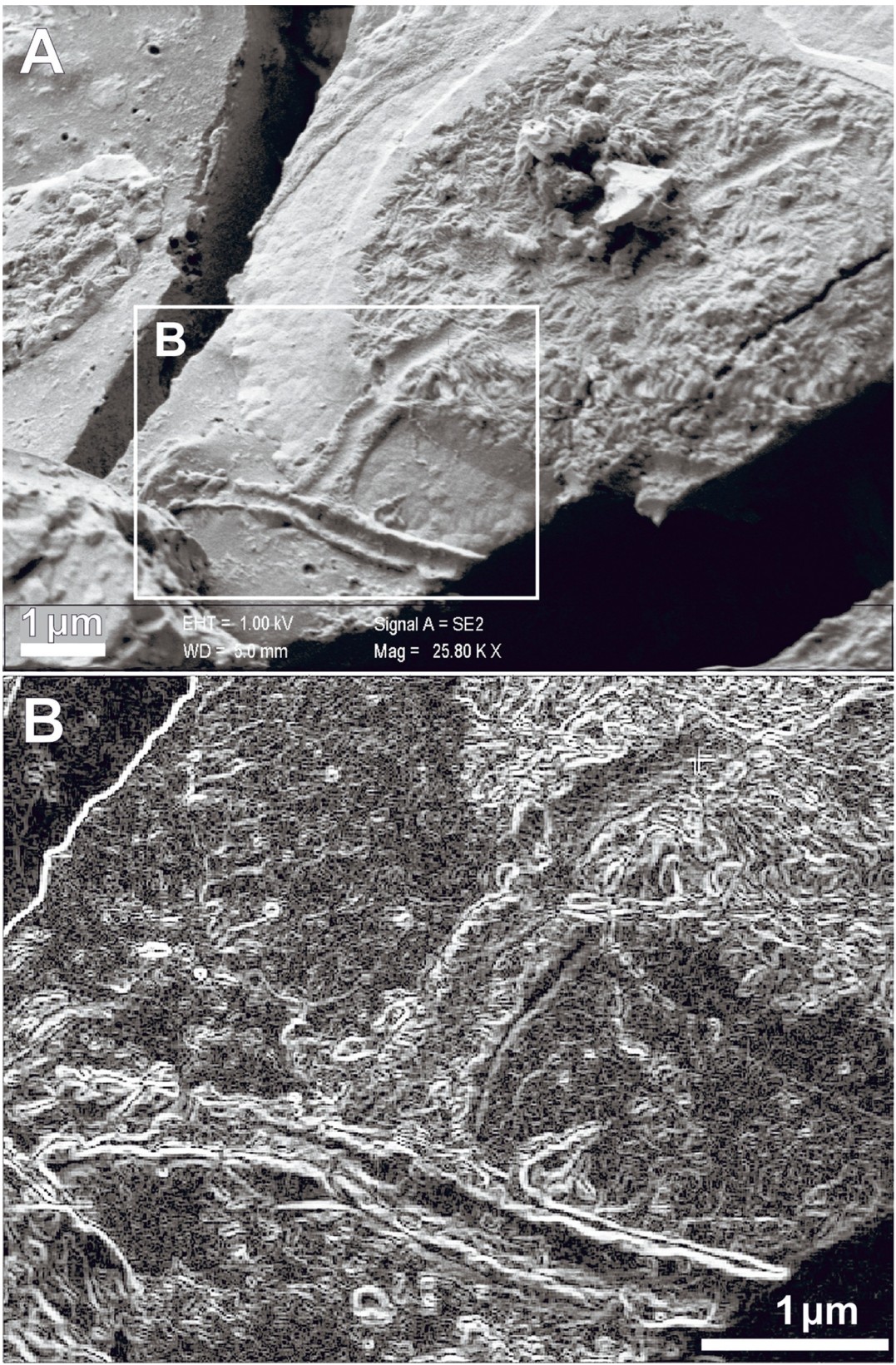

**Fig 4. Neuronal cell from the spinal cord.** A. A cell with an axon is bordered by the membrane. The cytoplasm is filled with filamentous structures; B. Neural network elaboration of SEM image B (white box). The cell cytoplasm shows a pile of tubular structures similar to microtubules (SEM, scale bars in micron).

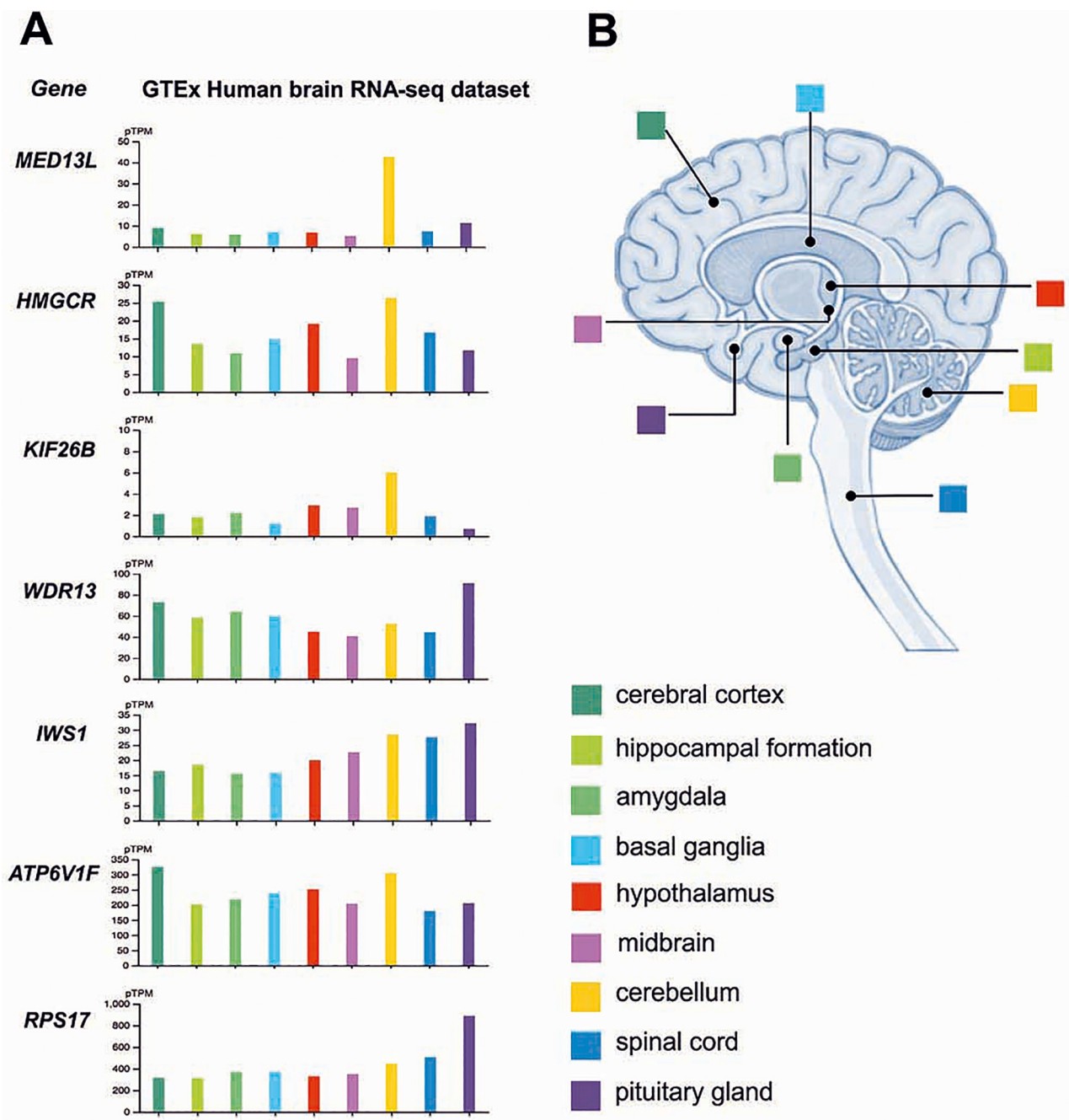

**Fig 5. Images of gene expression profiles of *ATP6V1F*, *HMGCR*, *KIF26B*, *IWS1*, *MED13L*, *WDR13* and *RPS17* obtained from the Allen Brain Atlas database (https://www.proteinatlas.org/).** A. The panel of diagrams shows the expression distribution across brain regions depicted by average expression for each region. Color coding is based on the brain subregions regions shown in B. Expression data were reported as mean pTPM (protein-coding transcripts per million) corresponding to mean values of the different individual samples for respective subregion; B. Midsagittal schematic of the different regions of the human brain.

**A**

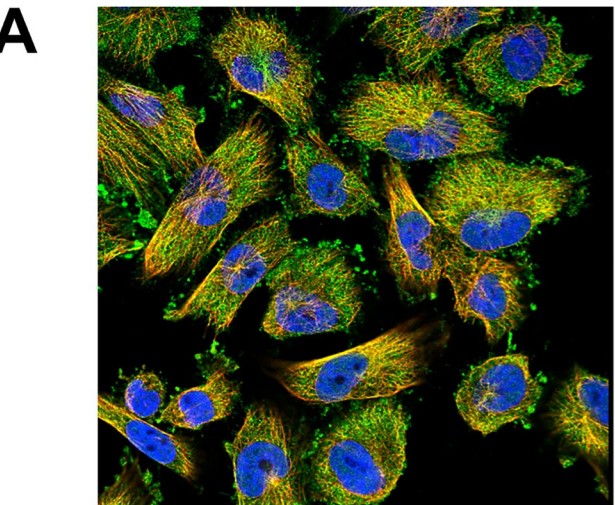

**B**

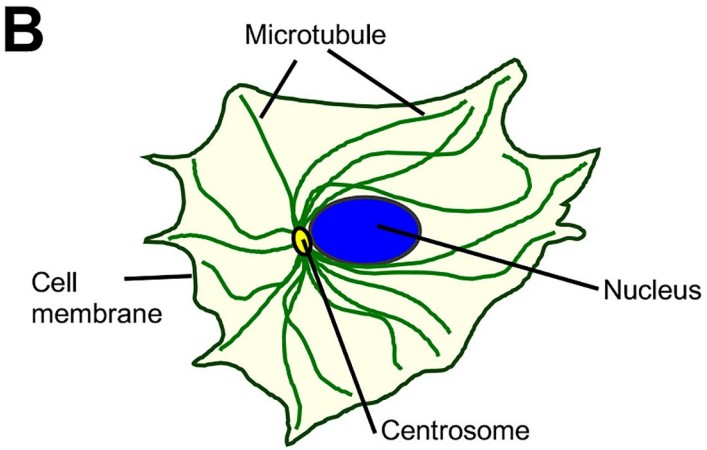

**C**

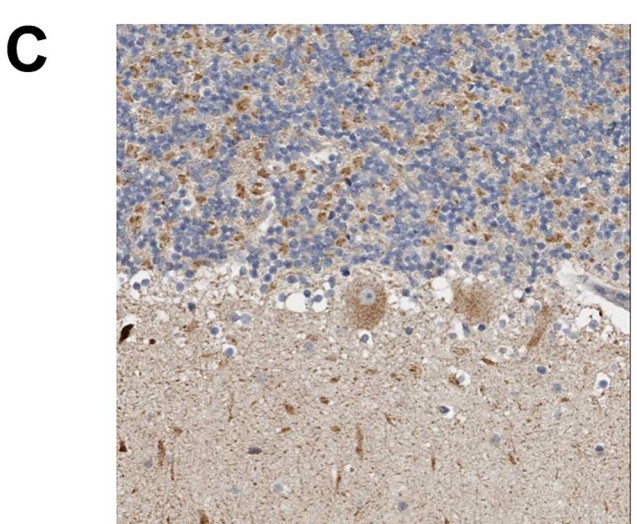

**Fig 6. Images of sub-cellular and cellular localization of KIF26B were obtained from the Allen Brain Atlas database (https://www.proteinatlas.org/).** A. Localization of KIF26B in microtubules and plasma membrane of human cell line U-2 OS; B. Cell component diagram; C. KIF26B signals in the human cerebellum (Purkinje cells and granular layer cells).

determination. The individual's age at death was determined using the degree of synostosis of the femoral head [55] and the stage of ectocranial suture closure [58].

## SEM imaging

The SEM images were taken from the vitrified tissue as it was, so as not to affect its integrity. Indeed, the samples being non-conductive, applying a metallic coating would have altered the surface characteristics. SEM imaging was performed by a Zeiss Sigma 300 Field Emission Scanning Electron Microscope (FESEM) (Department of Science, Università Roma Tre). Variable operational conditions (*e.g.*, accelerating voltage) are indicated within each image.

## Energy dispersive X-ray spectrometry (EDS)

The EDS analytical technique was used to obtain quantitative information on the composition of elements detected in the ROI by SEM. Elemental analyses were carried out with a Quantax EDS System, equipped with an XFlash energy-dispersive X-ray detector (Department of Science, Università Roma Tre). EDS operated at acceleration voltages of 15 kV for the measurements. EDS analysis was acquired using a 16 μs dwell time per pixel. The data were analyzed for the EDS spectrum with ESPRIT 2 software.

## Image processing and measurements

A morphological and morphometric analysis of the SEM images was carried out by a progressive segmentation of the original image through an unsupervised neural network. To better identify the evidenced structures it was necessary to develop a specific program related to fuzzy systems to evaluate their dimensions, *e.g.* areas, lengths, widths, "supporting skeleton", etc., with a minimum error (about 1/1000 pixels). The morphology of the scanned images was performed by applying the 2D Wavelet Transform, able to detect the presence of details usually lost by the filters used to reduce the noise of the signal, and post-processed by an unsupervised fuzzy neural network [34]. A region of interest (ROI) was identified before image acquisition. Geometrical measurements of the neuronal structures were performed through the segmentation of the ROI after filtering the background by means of the active contour detection process [35]. A home-developed program on Matlab® platform was used for image processing, both for the morphological process, and for morphometric analysis.

## Resource database

Expression data were obtained from the Allen brain atlas (https://atlas.brain-map.org/) and the Human Protein Atlas (HPA, www.proteinatlas.org) databases. It is a public online database providing an integrated overview of transcriptomic data and antibody-based protein profiling in all major CNS regions [43].

## Supporting information

**S1 File.**
(PDF)

## Acknowledgments

We thank the archaeological staff of the Archaeological Park of Herculaneum for their help in the field. We are particularly grateful to Sergio Lo Mastro for SEM-EDS analysis and imaging, performed during Covid-19 times. We would also like to thank Viola Desiato for helping in bibliographical research.

## Author Contributions

**Conceptualization:** Pierpaolo Petrone.

**Data curation:** Pierpaolo Petrone, Elena Vezzoli, Alessandra Pensa, Giuseppe Castaldo, Vincenzo Graziano, Emanuele Capasso, Giuseppe Quaremba, Maria Giuseppina Miano, Sergio Savino, Massimo Niola.

**Formal analysis:** Pierpaolo Petrone, Guido Giordano, Elena Vezzoli, Alessandra Pensa, Giuseppe Castaldo, Vincenzo Graziano, Francesco Sirano, Emanuele Capasso, Giuseppe Quaremba, Alessandro Vona, Maria Giuseppina Miano, Sergio Savino, Massimo Niola.

**Investigation:** Pierpaolo Petrone, Francesco Sirano.

**Methodology:** Pierpaolo Petrone.

**Supervision:** Pierpaolo Petrone.

**Validation:** Pierpaolo Petrone.

**Writing – original draft:** Pierpaolo Petrone, Guido Giordano, Elena Vezzoli, Giuseppe Castaldo, Giuseppe Quaremba, Maria Giuseppina Miano, Sergio Savino, Massimo Niola.

**Writing – review & editing:** Pierpaolo Petrone, Guido Giordano, Elena Vezzoli, Alessandra Pensa, Giuseppe Castaldo, Vincenzo Graziano, Francesco Sirano, Emanuele Capasso, Giuseppe Quaremba, Alessandro Vona, Maria Giuseppina Miano, Massimo Niola.

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
