## [Decision Letter · Decision Letter 0]

9 Sep 2020

PONE-D-20-24415

Preservation of Neurons in an AD 79 Vitrified Human Brain

PLOS ONE

Dear Dr. Petrone,

Thank you for submitting your manuscript to PLOS ONE. After careful consideration, we feel that it has a remarkable merit and just need slights changes to better describe parts of the samples as suggested by expert Reviewer #2. Therefore, we invite you to submit a revised version of the manuscript that addresses these very minor points raised during the review process and if I consider myself as clear enough, it would not be necessary any further round of revision.

Please ensure that your decision is justified on PLOS ONE’s publication criteria because your work can be considered as unique (at least unique up to date) and its impact is out of question.

We look forward to receiving your revised manuscript.

Kind regards,

Fernando de Castro

Academic Editor

PLOS ONE

Journal Requirements:

All necessary permits were obtained for the study of the human specimen.

The Ethics Committee for Biomedical Activities of the Azienda Ospedaliera Universitaria Federico II approved the study via written consent (Protocol approval number 101/17).

Please provide additional details regarding participant consent. In the ethics statement in the Methods and online submission information, please ensure that you have specified (1) whether consent was informed and (2) what type you obtained (for instance, written or verbal, and if verbal, how it was documented and witnessed). If your study included minors, state whether you obtained consent from parents or guardians. If the need for consent was waived by the ethics committee, please include this information.

3. We note you have included a table to which you do not refer in the text of your manuscript. Please ensure that you refer to Tables 1 & 2 in your text; if accepted, production will need this reference to link the reader to the Table.

4. Please include a caption for all of your figures

Reviewers' comments:

Reviewer's Responses to Questions

**Comments to the Author**

1. Is the manuscript technically sound, and do the data support the conclusions?

Reviewer #1: Yes

Reviewer #2: Partly

2. Has the statistical analysis been performed appropriately and rigorously? 

Reviewer #1: Yes

Reviewer #2: N/A

3. Have the authors made all data underlying the findings in their manuscript fully available?

Reviewer #1: Yes

Reviewer #2: Yes

4. Is the manuscript presented in an intelligible fashion and written in standard English?

Reviewer #1: Yes

Reviewer #2: Yes

5. Review Comments to the Author

Reviewer #1: This is an excellent analysis of the preservation of neural tissue by vitrification from a young male from Herculano. Authors provides detailed characterization of neurons and axons using SEM analysis.

Reviewer #2: This contribution is probably the most surprising article I have had the opportunity to review. Indeed, it is a difficult task to provide the authors with a well-argued opinion. Having a chance of inspecting a 2000-year-old vitrified CNS specimen of course is a very rare event. Here the authors took advantage of natural vitrification process enabling –for the first time if I am right- the accurate identification of neuronal structures at an unprecedented level of resolution. Taken together, this can be viewed as a “neuroarcheological” report.

My only concern is related to the method conducted for identifying neuronal processes from any other CNS structure like small blood vessels. As indicated in page 5, it seems that the main criteria is that identified tubular structures are smaller in diameter that blood vessels in the CNS. Although this is overall fine to me, I am wondering to what extent this will suffice. For instance, the structure shown in Figure 4 is here taken as a good example of non-myelinated axon arising from a spinal cord neuron. Although I am obviously not too much familiar with the conducted analytical procedure, I guess this could also be considered as a blood vessel of small caliber.

In summary, I guess this undoubtedly is a very rare example of neuroarcheology taken from a very rare event and of course of great interest from potential readers, likely representing a wide audience. Nevertheless and bearing in mind my background in neuromorphology, I guess the authors should make a greater effort in better explaining the criteria used for ensuring the neuronal origin of some of the observed structures.

6. PLOS authors have the option to publish the peer review history of their article (what does this mean?). If published, this will include your full peer review and any attached files.

Reviewer #1: No

Reviewer #2: No

---

## [Author Response · Author response to Decision Letter 0]

16 Sep 2020

Rebuttal letter – Ref.: PONE-D-20-24415

Title: Preservation of Neurons in an AD 79 Vitrified Human Brain

Response to the comments from the editor and reviewers

We wholeheartedly agree with comments from the editor and reviewers therefore we made a revision of the critical issues. In particular, following the suggestions of reviewer #2, we have treated in more depth and clarified the neuronal origin of the structure shown in figure 4, in order to avoid any possible misinterpretation concerning non-myelinated axons vs small blood vessels. 

As also suggested, we ensured to refer to Tables 1 & 2 in the text, and a caption for all of the figures has been added at the end of the manuscript. The four additional citations have also been added and renumbered within the text and bibliography as well. We have also improved the description of the databases used for the expression study, both in “Results” and “Materials and Methods” sections.

Reviewer #1: This is an excellent analysis of the preservation of neural tissue by vitrification from a young male from Herculano. Authors provides detailed characterization of neurons and axons using SEM analysis.

We thank the reviewer for his/her positive and insightful comments on the manuscript.

Reviewer #2: This contribution is probably the most surprising article I have had the opportunity to review. Indeed, it is a difficult task to provide the authors with a well-argued opinion. Having a chance of inspecting a 2000-year-old vitrified CNS specimen of course is a very rare event. Here the authors took advantage of natural vitrification process enabling ? for the first time if I am right- the accurate identification of neuronal structures at an unprecedented level of resolution. Taken together, this can be viewed as a "neuroarcheological" report.

My only concern is related to the method conducted for identifying neuronal processes from any other CNS structure like small blood vessels. As indicated in page 5, it seems that the main criteria is that identified tubular structures are smaller in diameter that blood vessels in the CNS. Although this is overall fine to me, I am wondering to what extent this will suffice. For instance, the structure shown in Figure 4 is here taken as a good example of non-myelinated axon arising from a spinal cord neuron. Although I am obviously not too much familiar with the conducted analytical procedure, I guess this could also be considered as a blood vessel of small caliber. In summary, I guess this undoubtedly is a very rare example of neuroarcheology taken from a very rare event and of course of great interest from potential readers, likely representing a wide audience. Nevertheless and bearing in mind my background in neuromorphology, I guess the authors should make a greater effort in better explaining the criteria used for ensuring the neuronal origin of some of the observed structures.

We thank the reviewer for his/her suggestion. We have now included two references (Müller et al., 2008; Schmid et al., 2019) in the main text showing that the smallest capillaries in the human brain have diameters ranging from 8 to 1 µm and a wall thickness of about 1 µm (Figure 1). 

Figure 1 . The typical diameters/wall thicknesses of blood vessel for humans and mice are given in micrometers. The smallest capillaries have diameters of a very few micrometers and a wall thickness of about one micrometer (from Müller et al., 2008).

In Figure 4 of our manuscript the mean diameter of a non-myelinated axon is 452·50 ∓ 20·50 nm, that is significantly inferior to the minimal dimension of the diameter of a capillary as reported in literature. Moreover, we can add two other important morphological details that can confirm our description. The first one is that this structure originates in fact from the changing curvature of the cellular membrane and it extends for several microns away from the cell body. To be clearer, here below we show a comparison between our SEM image (left) and a TEM image (right) of the axon initial segment (red arrows) of a cerebral cortical pyramidal neuron (Bender et al, 2012) that shows the same section that we found in our sample. Starting with this description we can assume that in our sample we have found a section of axon initial segment that is a non-myelinated axon (Palay et al., 1968). The last point that validates our description is that the wall of this structure is around 70 nm, that is far below the size of the wall thickness of the capillary ranging around 1 / 0.5 µm (Müller et al., 2008). 

Therefore, we have added the following sentences (page 6, lines 166-175 of Revised Manuscript with Track Changes):

"… diameter (nm, mean ∓ SD) (452·50 ∓ 20·50) (Fig 4A). Several evidence seems to confirm the neuronal origin of such structure: i. its diameter is significantly inferior to the minimal diameter of a capillary, the smallest one of which in the human brain have diameters ranging from 8 to 1 µm, and a wall thickness of about 1 µm [38,39]; ii. it originates from the cellular membrane and it extends for several microns away from the cell body; iii. its high similarity with the axon initial segment of a cerebral cortical pyramidal neuron [40]; iiii. the thickness of its wall (ca 70 nm) is far below the size of the wall thickness of a capillary (ca 1 to 0.5 µm) [38]. On this basis, we can assume that in our sample we have found a section of an axon initial segment which is a non-myelinated axon [41]".

Additional Bibliography

38. Müller B, Lang S, Dominietto M, Rudin M, Schulz G, Deyhle H et al. High-resolution tomographic imaging of microvessels. In Developments in X-Ray Tomography VI, Stuart R. Stock (ed), Proc. of SPIE Vol. 7078, 70780B, 2008; 0277-786X/08/$18 · doi: 10.1117/12.794157.

39. Schmid F, Barrett MJ, Jenny P, Weber B. Vascular density and distribution in neocortex. Neuroimage. 2019; 197:792-805.

40. Bender KJ, Trussell LO. The physiology of the axon initial segment. Ann Rev Neurosci. 2012; 35:249-265.

41. Palay SL, Sotelo C, Peters A, Orkand PM. The axon hillock and the initial segment. J Cell Biology. 1968; 38(1):193-201.

---

## [Editor Report · Decision Letter 1]

18 Sep 2020

Preservation of Neurons in an AD 79 Vitrified Human Brain

PONE-D-20-24415R1

Dear Dr. Petrone,

We’re pleased to inform you that your manuscript has been judged scientifically suitable for publication and will be formally accepted for publication once it meets all outstanding technical requirements.

Kind regards,

Fernando de Castro

Academic Editor

PLOS ONE
---

## [Editor Report · Acceptance letter]

24 Sep 2020

PONE-D-20-24415R1 

Preservation of Neurons in an AD 79 Vitrified Human Brain 

Dear Dr. Petrone:

I'm pleased to inform you that your manuscript has been deemed suitable for publication in PLOS ONE. Congratulations! Your manuscript is now with our production department. 

Kind regards, 

on behalf of

Dr. Fernando de Castro 

Academic Editor

PLOS ONE